# Characterization of a Novel RNA Virus Causing Massive Mortality in Yellow Catfish, *Pelteobagrus fulvidraco*, as an Emerging Genus in *Caliciviridae* (*Picornavirales*)

Wenzhi Liu,[a,b] Mingyang Xue,[a] Tao Yang,[c] Yiqun Li,[a] Nan Jiang,[a] Yuding Fan,[a] Yan Meng,[a] Xiaowen Luo,[a] Yong Zhou,[a] Lingbing Zeng[a]

[a]Yangtze River Fisheries Research Institute, Chinese Academy of Fishery Sciences, Wuhan, People's Republic of China
[b]College of Fisheries, Huazhong Agricultural University, Wuhan, People's Republic of China
[c]Animal Health Research Institute, Tongwei Co., Ltd., Chengdu, People's Republic of China

**ABSTRACT**  An emerging disease in farmed yellow catfish (*Pelteobagrus fulvidraco*) causing massive mortality broke out in 2020 in Hubei, China. Histopathological examination indicated significant changes in kidneys and spleens of diseased fish. Electron microscopy revealed large numbers of viral particles in the kidneys and spleens. These particles were spherical with a diameter of approximately 35 nm. By using RNA sequencing and rapid identification of cDNA ends, the full nucleotide sequence of the virus was identified. The viral genome comprises 7,432 bp and contains three open reading frames sharing no nucleotide sequence similarity with other viruses; however, the amino acid sequence partially matched that of the nonstructural (NS) proteins from viruses in the order *Picornavirales*. Combined with the phylogenetic analysis, the conserved amino acid motifs and the domains of the viral genome predict a genome order typical of a calicivirus. Therefore, this virus was tentatively named yellow catfish calicivirus (YcCV). Cell culture showed that YcCV could cause a cytopathic effect in the channel catfish kidney cell line (CCK) at early passages. In artificial infection, this virus could infect healthy yellow catfish and led to clinical symptoms similar to those that occurred naturally. *In situ* hybridization analysis detected positive signals of the virus in kidney, spleen, liver, heart, and gill tissues of diseased fish. This study represents the first report of calicivirus infection in yellow catfish and provides a solid basis for future studies on the control of this viral disease.

**IMPORTANCE**  Caliciviruses are rapidly evolving viruses that cause pandemic outbreaks associated with significant morbidity and mortality globally. A novel calicivirus identified from yellow catfish also causes substantial mortality. Using an RNA sequencing (RNA-seq) and rapid amplification of cDNA ends (RACE) method, the full nucleotide sequence was identified and characterized, and this virus was tentatively named yellow catfish calicivirus (YcCV). A nucleotide sequence similarity search found no match with other viruses, and an amino acid sequence comparison indicated approximately 23.3% amino acid homology with the viruses in the order *Picornavirales*. These findings may represent a new avenue to explain virus evolution and suggest a need to further study the pathogenesis of calicivirus and characterize possible interactions among interspecific viruses in the aquaculture environment.

**KEYWORDS**  yellow catfish, *Pelteobagrus fulvidraco*, *Picornavirus*, *Calicivirus*, characterization, phylogenetic analysis

The yellow catfish, *Pelteobagrus fulvidraco*, belonging to the family *Bagridae*, is an economically important freshwater species in China and other Asian countries. This fish is prized for its flesh quality, rapid growth, and high economic value (1). Recently, yellow catfish aquaculture has developed rapidly in China, and in 2020, the annual

**Address correspondence** to Lingbing Zeng, zlb@yfi.ac.cn, or Yong Zhou, zhouy@yfi.ac.cn.

The authors declare no conflict of interest.

production of the fish reached 0.57 million tons (2). However, alongside the rapid development of the industry, diseases of this fish have become a great concern (3–7). In 2020, a severe disease of yellow catfish occurred in the city of Qianjiang, in the province of Hubei. Meanwhile, the same clinical symptoms of diseased fish were found in other cities in Hubei and in other provinces, such as Sichuan, Zhejiang, and Guangdong, etc. The diseased fish were characterized by hemorrhage on the head, mouth, lower jaw, and fin bases, and the cumulative mortality could exceed 90%. Overall, the epidemic features were quite different from those of diseases previously reported in yellow catfish. Moreover, this disease could not be controlled by water disinfection or oral administration of antibacterial drugs.

Identifying potential pathogens in host can be carried out conveniently and efficiently by using next-generation sequencing (8). In this method, RNA is amplified randomly and subjected to deep sequencing to generate transcript data. *In silico* translation of the transcripts provides protein sequences that can be compared with all known pathogenic sequences in databases. Thus, the present study aimed to characterize the causative pathogen in diseased yellow catfish using next-generation sequencing. Moreover, the pathogen was further confirmed by using electron microscopy, sequence characterization, reverse transcription-PCR (RT-PCR), animal experiments, *in situ* hybridization, and phylogenetic analysis. This study represents the first report of calicivirus infection in yellow catfish and provides a solid basis for the further study on the infection mechanism of this virus and the control method for the disease in future.

## RESULTS

**Disease and pathological features.** In April 2020, a disease initially broke out in cultured yellow catfish in the city of Qianjiang, in the province of Hubei, China, which displayed high mortality (>90%). The disease was highly contagious while the water temperature in ponds ranged from 20°C to 24°C and caused death of yellow catfish of all sizes. The diseased fish hung head up in water, exhibiting disoriented behaviors (Fig. 1A). Clinical signs included hemorrhages on the head, mouth, lower jaw, and fin bases (Fig. 1B). The spleen was dark and enlarged, and the kidney was hemorrhagic (Fig. 1C and D). No parasite was found in the gill of diseased fish by microscopic examination. Brain heart infusion agar plates were used for bacterial isolation from the liver and kidney of moribund fish, and no bacterial infection was confirmed.

In diseased fish, histopathological analysis indicated that the kidney and spleen appeared most severely affected, showing extensive necrosis, vacuolation, and disappearance of tissues (Fig. 1E and F). Spleen appeared severely affected, showing different levels of extensive necrosis and various-sized vacuolation (red arrow) (Fig. 1E). Kidney edema was accompanied by moderate to heavy infiltration of lymphocytes (Fig. 1F, blue arrow). In addition, the nucleus was condensed and marginated in the cells of the infected kidney (Fig. 1F, black arrow).

**RNA sequencing and virus isolation.** The clinical symptoms of this emerging disease are quite different from those of diseases reported previously in yellow catfish, implying that the causative agent was uncharacterized. By transmission electron microscopy observation, a large number of spherical viral particles were revealed in the cytoplasm of cells in kidney and spleen of diseased fish. The virus particle was approximately 35 nm in diameter (Fig. 2A and B). Therefore, an RNA sequencing (RNA-seq) approach was performed using kidney and spleen tissues of diseased fish. Nonspecific short sequences of the host, plants, bacteria, parasites, and fungi were screened out before assembly of the transcripts (Fig. 2C). The results showed that many of the unassigned sequences represented yellow catfish genes and low-abundance RNA contaminants (Fig. 2D and E). A total of 4,359 reads were generated and assembled, of which 65% were found to be homologous with members of the order *Picornavirales* according to BLASTp analysis using conceptual translations of the transcript sequences. No other genes that resembled those from replication-competent viruses were found. Manual alignment of viral sequences produced a 6,763-bp fragment that encodes a nonstructural (NS) protein. Amino acid sequence comparison indicated approximately

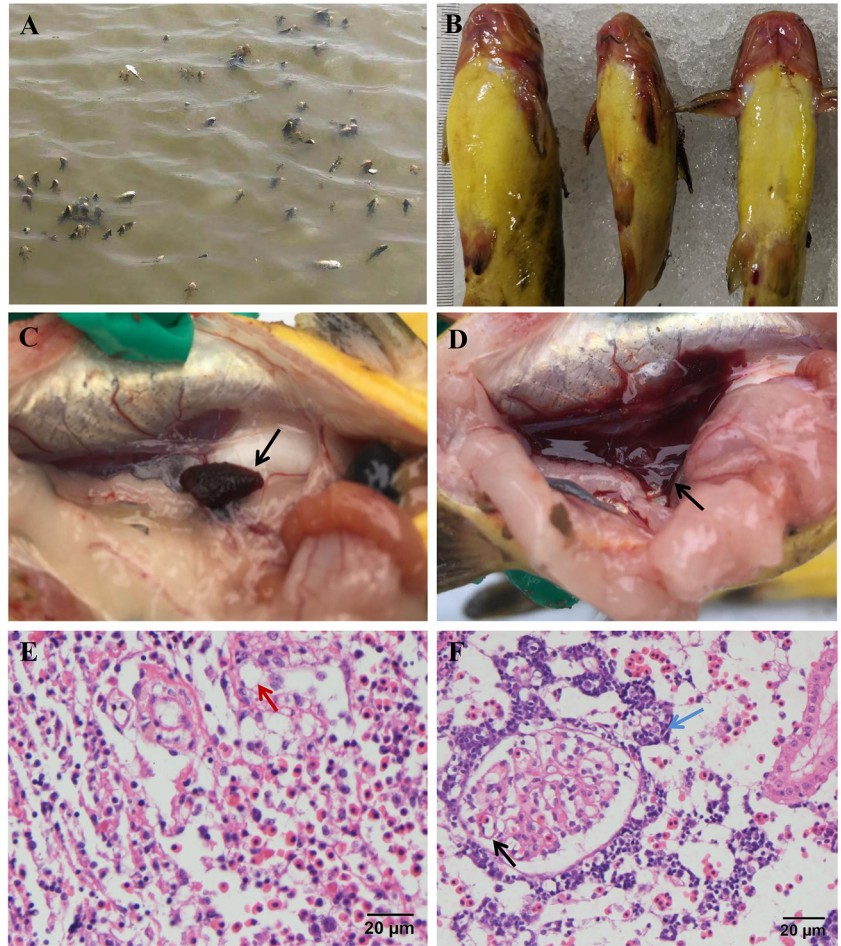

**FIG 1** Clinical signs of diseased yellow catfish. (A) Diseased yellow catfish hanging head up in water and exhibiting disoriented behavior; (B) diseased yellow catfish showing hemorrhages on head, mouth, lower jaw, and fin base; (C) spleen of diseased fish, showing a dark color; (D) kidney showing severe hemorrhagic and necrosis; (E) diseased fish spleen showing various-sized vacuolation (red arrow) by H&E staining; (F) diseased fish kidney showing edema and a moderate to heavy infiltration of lymphocytes (blue arrows) and condensed and marginated nuclei of glomerulus cells (black arrow) by H&E staining.

23.3 to 28.9% amino acid sequence similarity with NS proteins from the order *Picornavirales*. According to The International Committee on Taxonomy of Viruses guidelines, different genera of the order *Picornavirales* shared approximately <65% amino acid identity to the NS or VP1 proteins; therefore, the novel virus was speculated to be a member of the order *Picornavirales*.

Tissue homogenates for the initial virus isolation were inoculated onto channel catfish kidney (CCK), channel catfish ovary (CCO), grass carp ovary (GCO), epithelioma papilloma cyprinid (EPC), rainbow trout gonadal (RTG-2), fathead minnow (FHM), gibel carp brain (GiCB), and Chinese rice field eel kidney (CrEK) cell lines. Among them, CCK was the most susceptible after three consecutive blind passages of virus culture (Fig. 2G), and no cytopathic effect was observed in uninfected CCK cells (Fig. 2F). Electron microscopy confirmed that the viral particles in cells were similar to those in the naturally infected fish (Fig. 2H and I).

**Genome sequencing analysis of yellow catfish calicivirus (YcCV).** To obtain the full viral genome sequence, overlapping PCR and rapid amplification of cDNA ends (RACE) were carried out using a gene-specific primer designed according to the partial viral sequence obtained from the RNA-seq approach (Fig. 3A to C). The complete genome of this virus comprised 7,432 bp. The complete RNA sequence was used to measure the

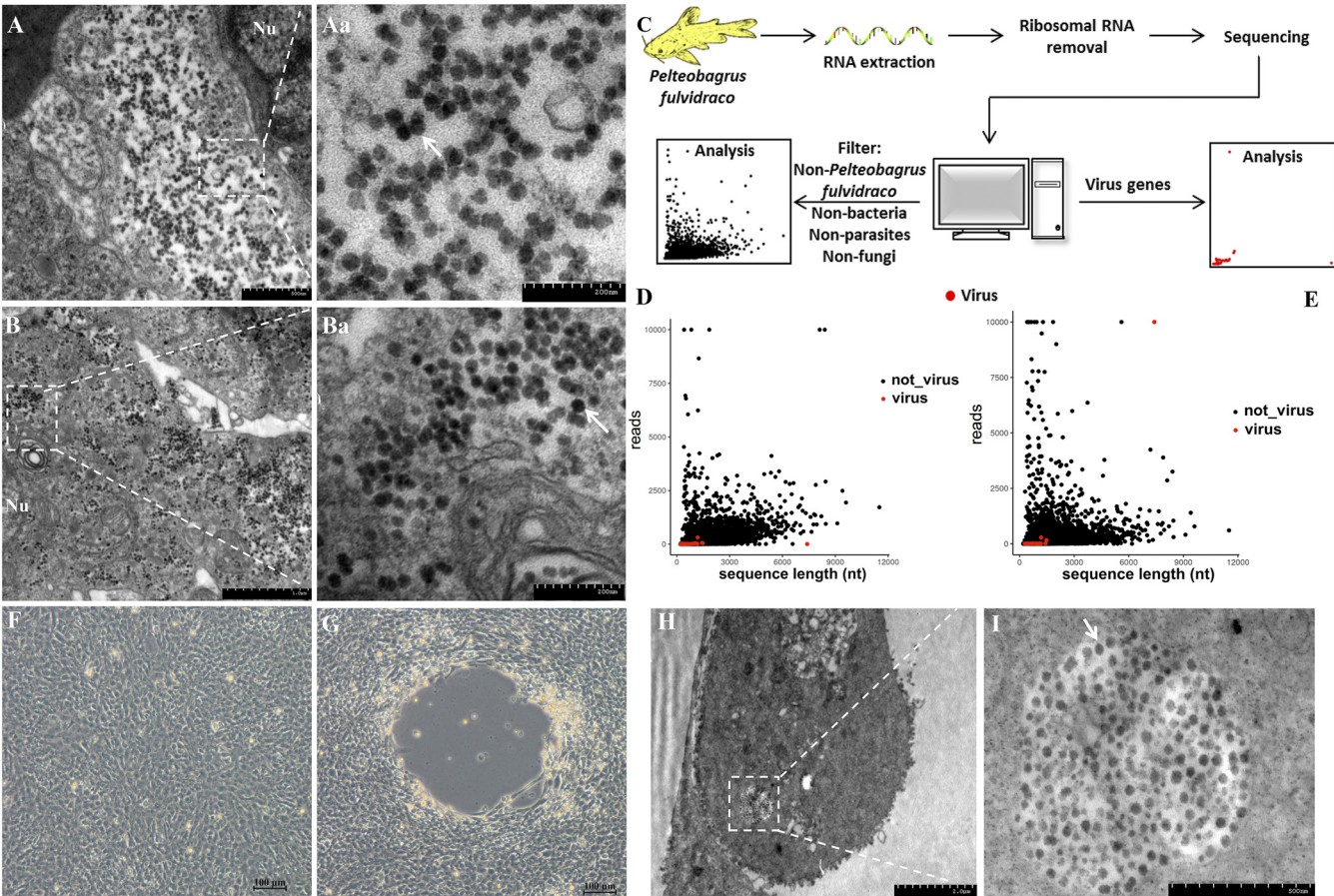

**FIG 2** Identification of the causative pathogen in diseased yellow catfish using RNA-seq, transmission electron microscopy, and virus isolation. (A) Kidney. Viral particles were observed in the cytoplasm. Nu, nucleus. Bar = 1 $\mu$m. (Aa) Higher magnification of the area bounded by the white rectangle in panel A showing magnified virus particles (white arrow). Bar = 200 nm. (B) Spleen. The mature viruses clustered in the cytoplasm near the cell nucleus. Bar = 1 $\mu$m. (Ba) Higher magnification of the area bounded by the white rectangle in panel B showing magnified virus particles (white arrow). Bar = 500 nm. (C) Workflow for RNA sequencing of kidney and spleen samples from diseased yellow catfish. (D and E) Viral mRNA sequence identification in kidney and spleen samples from diseased fish. Numbers and lengths of nonviral mRNA sequences (black solid dot) and predicted viral mRNA sequences (red solid dot) are shown. (F) Normal CCK cell. (G) CCK cell infected with the third passage of YcCV at 7 days postinfection. Typical CPE is visible. Bar = 100 $\mu$m. (H) Virus particles in CCK cells. Bar = 2.0 $\mu$m. (I) Higher magnification of the area bounded by the white rectangle in panel H showing magnified virus particles (white arrow). Bar = 500 nm.

coverage of the virus mRNA across the viral genome using the kidney sample (Fig. 3D). The results showed that the expression of the NS protein, capsid protein, and three open reading frame (ORF) proteins (ORF1 to ORF3) likely proceeded via alternative splicing in diseased yellow catfish. Sanger sequencing of the RT-PCR products generated from a diseased kidney sample was used to confirm the fidelity of the alternative splicing process (Fig. 3D). These three ORFs are not overlapping, and a 5′ untranslated region (UTR) (45 bp) and a 3′ poly(A) tail (58 bp) were identified at the 5′ and 3′ termini of the viral genome, respectively (GenBank accession no. MZ065194) (Fig. 3A).

The ORF1 gene of this virus encodes an NS protein, characterized by a polyprotein domain. A BLASTX nucleotide sequence similarity search revealed no match with sequences from other viruses. However, the conceptually translated amino acid sequence was approximately 23.3 to 28.9% similar to NS protein from viruses in the order *Picornavirales*, including the families *Caliciviridae* (GenBank accession no. AQQ78883), *Dicistroviridae*, and *Picornaviridae*, and a recently discovered picorna-like virus (unclassified *Picornavirales*) (9–13). The virus NS protein (1,671 amino acids [aa]) contained the NTPase/helicase motif $^{376}$GXPGXGKT$^{383}$ and the RNA-dependent RNA polymerase (RdRp) motif $^{1524}$YGDD$^{1527}$. Another two RdRp motifs, GLPGS and DYXXWDST, were present as $^{1479}$GMPSG$^{1483}$ and $^{1421}$DYSGFDST$^{1428}$, respectively (Fig. 3E). ORF2

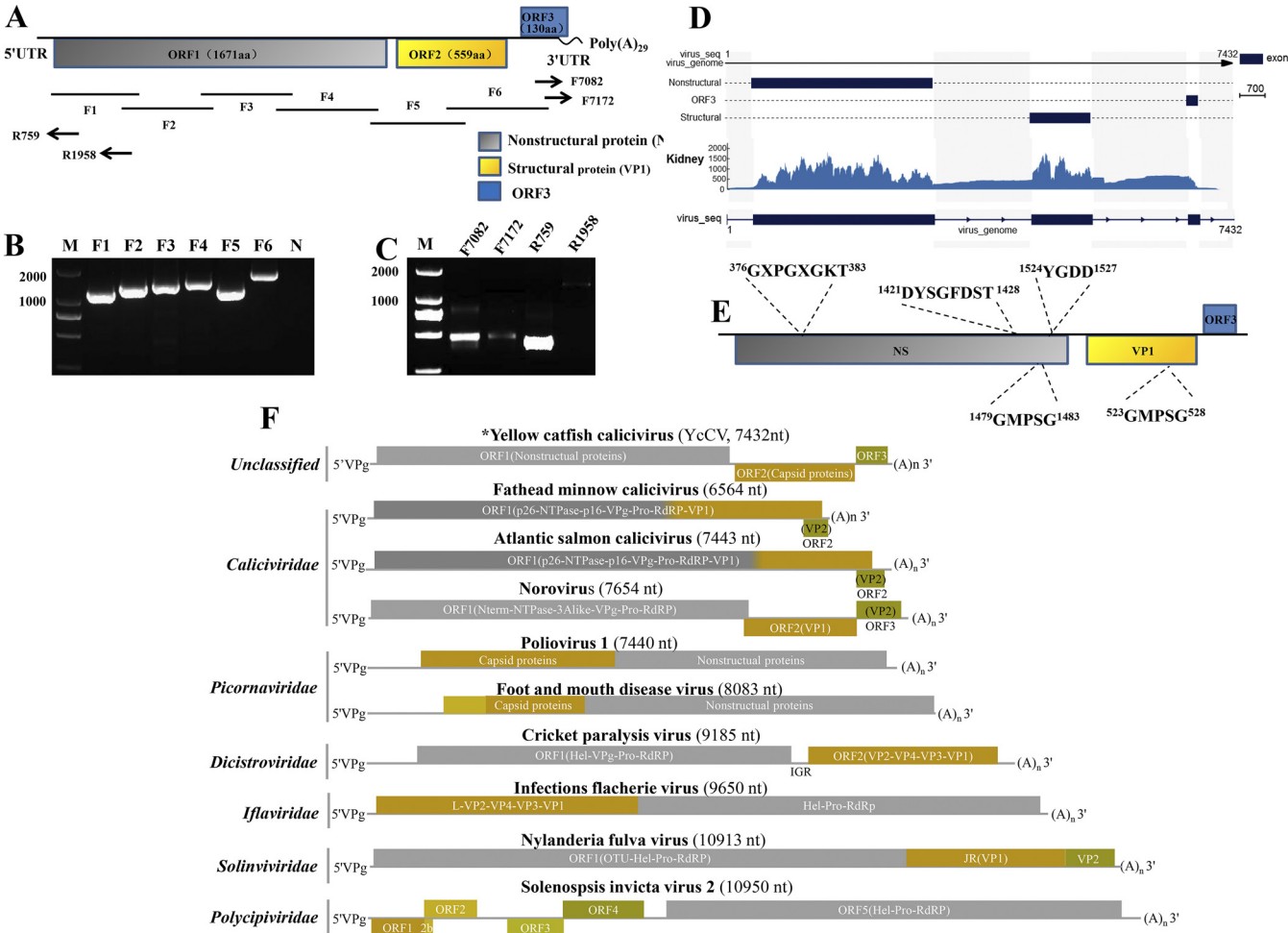

**FIG 3** Genome sequencing analysis of YcCV. (A) The assembled genome YcCV. The genome is composed of three ORFs (NS, VP1, and ORF3), a 5′ UTR at the 5′ terminus, and a 3′ poly(A) tail at the 3′ terminus. (B) Overlapping RT-PCR was carried out with six pairs of primers (YcCV-F1/YcCV-R1 to YcCV-F6/YcCV-R6) to obtain the main sequences of YcCV. (C) 5′ and 3′ RACE amplification of the full-length YcCV genomic sequence using the primers YcCV-rF1, -rF2, -rR1, and -rR2. (D) mRNA reads from the kidney tissue that mapped to the complete viral genome were confirmed by using RACE, RT-PCR from kidney tissue, and Sanger sequencing. Blue mapping indicates major splice donor and acceptor sites. The lower diagram shows the genomic structure of the YcCV sequence from the kidney of yellow catfish. Blue boxes indicate the nonstructural protein (NS), capsid protein (VP1), and ORF3 protein coding regions, and arrowheads show direction of YcCV transcription. (E) The location of conserved domains for an RNA helicase, RdRp, and capsid protein covering the regions found in YcCV. (F) Diagram comparing the genomic structures of members in the order *Picornavirales* with that of the calicivirus isolated from yellow catfish. An asterisk indicates YcCV derived from the present study.

comprises 562 aa and was significantly shorter than ORF1. The complete protein sequence of ORF2 shares no amino acid sequence similarity with virus protein sequences in the NCBI database. However, the converted structural motif GMPSG of a capsid protein at positions 523 to 528 was identified (Fig. 3E). According to the genome organization of caliciviruses in the order *Picornavirales*, the ORF2 protein of this virus was predicted to be a structural protein (capsid protein). Last, ORF3 was identified to be transcribed in the same direction as ORF2 but from a different reading frame. The conceptually translated ORF3 protein comprised 130 aa and showed no similarity to other proteins translated from *Picornavirales* genomes upon BLASTp analysis (Fig. 3F). The genomic structure of the virus's ORF usage and its gene order were similar to those of *Norovirus*, *Recovirus*, and *Vesivirus* in the family *Caliciviridae*, even though the virus has some sequence similarity to the families *Dicistroviridae* and *Picornaviridae* in the order *Picornavirales* (Fig. 3F). In addition, searching for conserved protein domains using the Conserved Domain Database in NCBI also showed conserved domains for an RNA helicase, an RdRp, and capsid protein covering the regions where the conserved amino acid motifs were found, together with a domain for a separate capsid protein and minor ORF3 (Fig. 3A). This analysis confirms that the NS protein, capsid protein, and ORF3 protein of the virus are

organized in a similar order to other caliciviruses when compared with the members of the order *Picornavirales* (Fig. 3F). Therefore, according to the genomic structure analysis, length, viral particle size, gene order, and typical domains, the virus was tentatively named yellow catfish calicivirus (YcCV), being a calicivirus in the order *Picornavirales*.

**Detection, transmission, and distribution of YcCV.** The NS protein sequence was used to design primers to detect the virus in diseased samples (Fig. 4A), which amplified a PCR fragment of 523 bp (Fig. 4B). Healthy yellow catfish were challenged with bacterium-free tissue homogenates derived from naturally YcCV-infected yellow catfish mainly showing hemorrhagic symptoms that were similar to those found in naturally diseased fish. Samples from dead or moribund fish in challenged groups generated the predicted size of PCR fragment, while in samples from the mock-infected group, no amplification was observed (Fig. 4C). In the experimental infection group, mortality was as high as 90% at 10 days postinfection (dpi) (Fig. 4D), while in the mock-infected group, all the fish remained asymptomatic. In addition, the kidney and spleen tissues of yellow catfish from the infected group were used to detect YcCV by *in situ* hybridization, and the location of the probe in the YcCV genome is shown in Fig. 4E. The results showed positive signals for YcCV kidney and spleen cells from infected fish (Fig. 4G and I), but no signals were detected from yellow catfish in the control group (Fig. 4F and H). These results clearly verified YcCV as the etiological agent responsible for the emerging disease in yellow catfish farms.

YcCV was assessed in gill, brain, intestine, heart, kidney, spleen, and liver tissues of naturally infected yellow catfish using RT-PCR. Quantitative real-time PCR was used to determine viral genome copies. RT-PCR detection showed that, except for the intestines, all the other tissues were YcCV positive (Fig. 4J). The quantity virus genome copies in infected tissues ranged from $3.0 \times 10^{5.05 \pm 0.17}$ to $2.5 \times 10^{7.07 \pm 0.25}$ copies/mg (Fig. 4K). Tissue samples from diseased yellow catfish were tested for the presence of YcCV using NS-targeted RT-PCR (YcCV-F/R). Samples from nine cities in two provinces showed positivity for YcCV, including Hubei (Qianjiang [14/32; 43.8%], Jingmen [5/18; 27.8%], Jiayu [3/12; 25.0%], Wuhan [5/21; 23.8%], Xiantao [8/23; 34.8%], Yichang [4/16; 25.0%], Zhijiang [10/21; 47.6%], and Jingzhou [2/17; 11.8%]) and Sichuan (Leshan [4/12; 33.3%]). The positive whole viral genome sequences from nine cities were sequenced, and the genes showed 94.6% to 99.2% nucleotide identity (Fig. 4L).

**Fluorescence *in situ* hybridization detection of YcCV in naturally infected yellow catfish.** The localization of viral RNA in naturally infected yellow catfish was investigated by fluorescence *in situ* hybridization (FISH) (Fig. 5; also, see Fig. S1 in the supplemental material). Bright green fluorescence, indicating positive hybridization signals, was observed in the gill, liver, heart, spleen, and kidney; however, the signal intensity varied. The kidney and spleen showed the strongest signals (Fig. 5A and C). The negative-control samples showed no specific fluorescent signals (Fig. 5Ba and Da). The cytoplasm of the infected cells showed the maximum fluorescence intensity (Fig. 5Aa and Ca).

**Phylogenetic analysis and classification.** Phylogenetic analysis was based on the putative amino acid sequences of complete NS proteins of seven families in the order *Picornavirales*, including *Caliciviridae*, *Dicistroviridae*, *Iflaviridae*, *Picornaviridae*, *Polycipiviridae*, *Secoviridae*, and *Solinviviridae* (Fig. 6). The sequence similarities of the deduced YcCV nonstructural protein with NS proteins from *Caliciviridae*, *Dicistroviridae*, and *Picornaviridae* were 23.3 to 28.9%, suggesting a possible unclassified novel virus family within the order *Picornavirales*. Phylogenetic analysis showed that YcCV was close to the *Solinviviridae* and *Caliciviridae* and closer to *Nyfulvavirus* in the family *Solinviviridae*; however, the YcCV NS was found to possess ~27.23% amino acid similarity to other caliciviruses but no similarity to the viruses in *Solinviviridae* using NCBI BLASTp analysis. In addition, monophyly of the *Solinviviridae* within the larger picorna/calici-like group is not completely certain, and it is possible that the *Solinviviridae* form a sister group to the *Caliciviridae*, although the phylogenetic clustering is inconclusive because of the difficulty in resolving tree topologies at this depth (14, 15). In our study, the viral morphology, viral particle size, viral nucleic acid

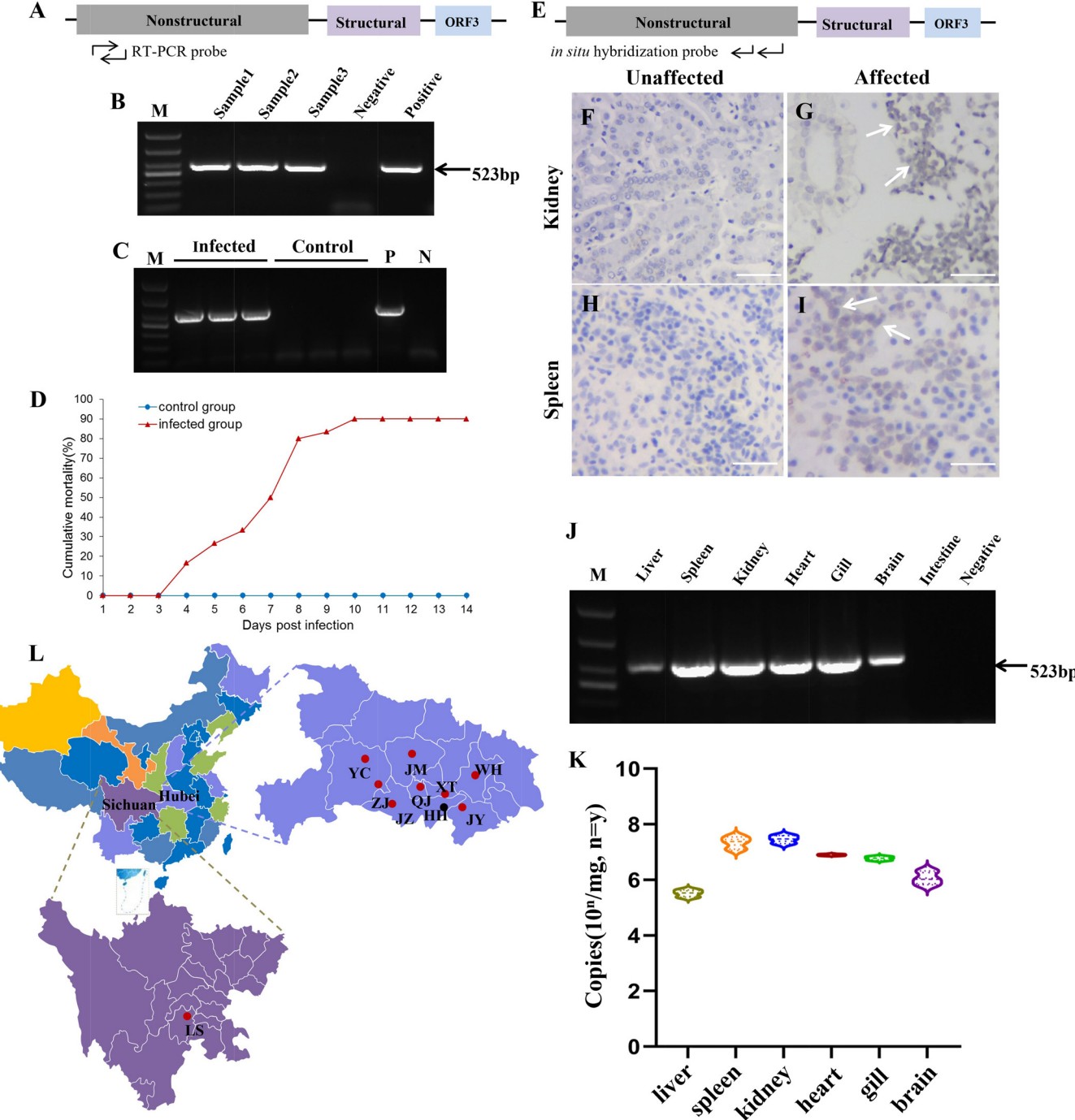

**FIG 4** Detection, transmission, and distribution of YcCV in diseased yellow catfish. (A) Locations of RT-PCR primers on the YcCV genome. (B) Detection of YcCV in naturally infected yellow catfish using RT-PCR. Lane M, 1,000-bp DL1000 DNA ladder; lanes 1 to 3, results for kidney tissue samples taken from three naturally infected yellow catfish (samples 1 to 3); lane 4, negative control; lane 5, positive control. (C) RT-PCR detection of YcCV in the experimental group and the control group (i.p. injection of Dulbecco's PBS). Lane M, DL1000 1,000-bp DNA ladder. Lanes 1 to 3, results for kidney tissues samples taken from three naturally infected yellow catfish; lanes 4 to 6, results for three mock-infected yellow catfish; lane 7, positive control (P); lane 8, negative control (N). (D) Mortality of yellow catfish in the experimental group and the control group. A homogenate of diseased fish tissues was prepared and filtered through a 0.22-$\mu$m filter (10 to 13 cm in length), and 0.5 mL was injected i.p. to challenge 30 fish in the experimental group. (E) Location of the *in situ* hybridization probe for YcCV. (F) Kidney samples from mock-infected yellow catfish. (G) Kidney samples with positive signals in infected yellow catfish (white arrow) using *in situ* hybridization. (H) Spleen samples from mock-infected yellow catfish. (I) Spleen samples with positive signals in infected yellow catfish (white arrow). (J) Distribution of YcCV in gill, brain, intestine, heart, kidney, spleen, and liver tissues of naturally infected yellow catfish. Lane M, 1,000-bp DL1000 DNA ladder; lane 1, liver; lane 2, spleen; lane 3, kidney; lane 4, heart; lane 5, gill; lane 6, brain; lane 7, intestine; lane 8, negative control. (K) Numbers of copies of YcCV in seven different tissues from naturally infected yellow catfish. Data are means and standard errors of the means for 3 independent replicates. (L) Locations of diseased yellow catfish samples collected in Hubei Province and Sichuan Province. Circles show sampling sites of yellow catfish; red indicates samples that were positive for YcCV, and black indicates negative samples.

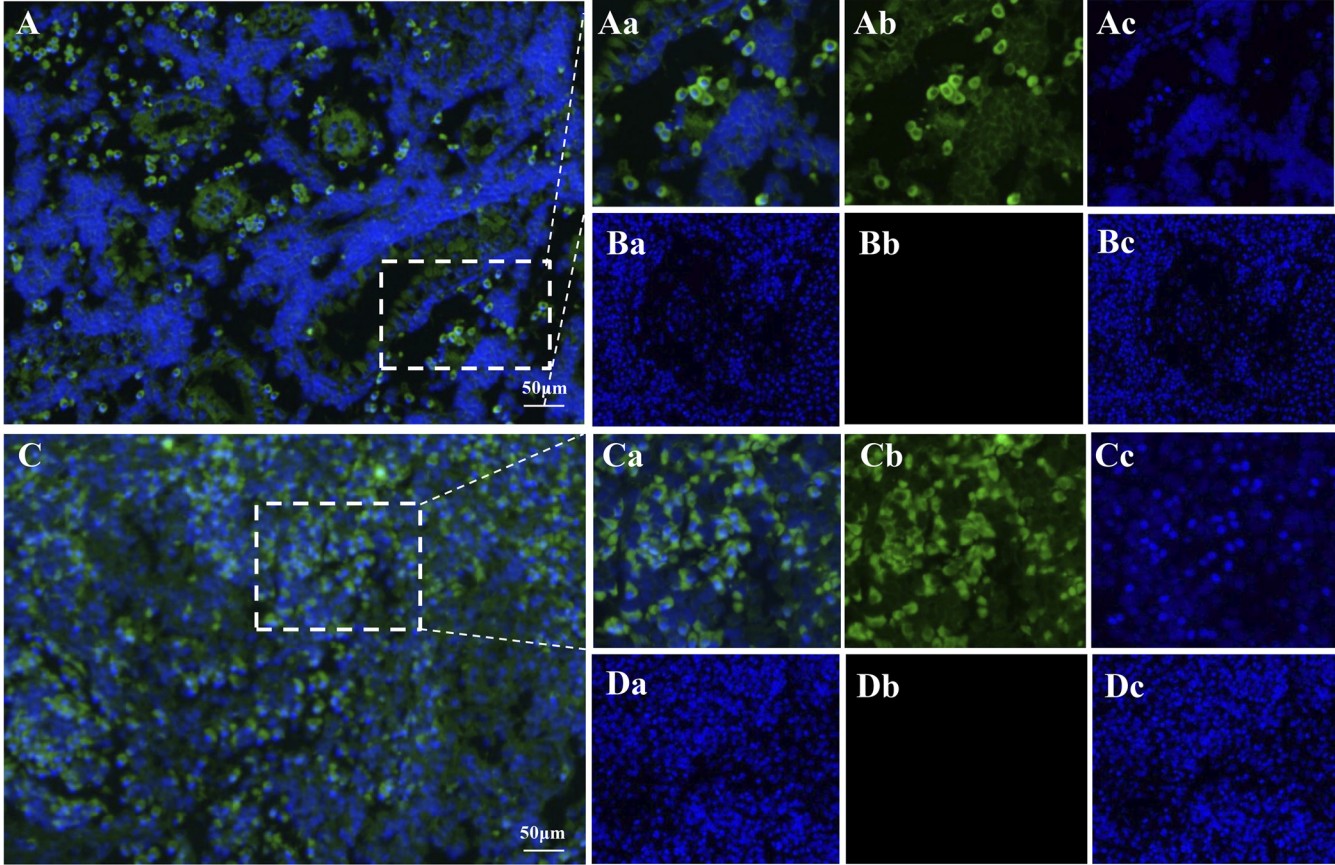

**FIG 5** FISH detection of YcCV in naturally infected yellow catfish. (A and C) FISH hybridization in YcCV-infected kidney and spleen cells, respectively; (Aa, Ca) detection of positive signals in kidney and spleen cells, respectively; (Ba, Da) absence of signal in healthy kidney and spleen cells. Arrows show positive signals.

length, genome organization, and host range suggest that YcCV might belong to a new genus in the family *Caliciviridae*.

## DISCUSSION

Yellow catfish, *Pelteobagrus fulvidraco*, is an economically important fish that is cultured widely in Southeast Asian countries, especially China (1). In recent years, with the rapid development of the yellow catfish farming industry in China, several diseases of this fish leading to economic losses have been well documented (3–7). However, an emerging severe disease of yellow catfish which is highly contagious broke out in farmed yellow catfish in 2020 and threatened the sustainable development of the yellow catfish farming industry in China. In the present study, the causative agent of this yellow catfish disease was identified as a novel calicivirus, tentatively named yellow catfish calicivirus (YcCV). Members of the family *Caliciviridae* in the order *Picornavirales* are small, nonenveloped, and icosahedral-symmetry viruses with a diameter of 27 to 40 nm (16). This family has been detected in a wide range of vertebrates and is presently classified into 11 genera; seven of them infect mammals (*Vesivirus*, *Valovirus*, *Sapovirus*, *Recovirus*, *Nebovirus*, *Norovirus*, and *Lagovirus*), two infect birds (*Bavovirus* and *Nacovirus*), and two infect fish (*Minovirus* and *Salovirus*) (17–21).

The calicivirus genome is a linear positive-sense RNA molecule 6.4 to 8.5 kb in length with two or three major ORFs (16). ORF1 encodes a polyprotein that is processed proteolytically into at least six NS proteins: the N-terminal protein (Nterm), the 2C-like NTPase, the 3A-like protein, VPg, the 3C-like protease (Pro), and the 3D-like RdRp. Members of the family *Caliciviridae* encode a major structural capsid protein, VP1, in the same direction on the viral genomic sequence as the NS protein, while in

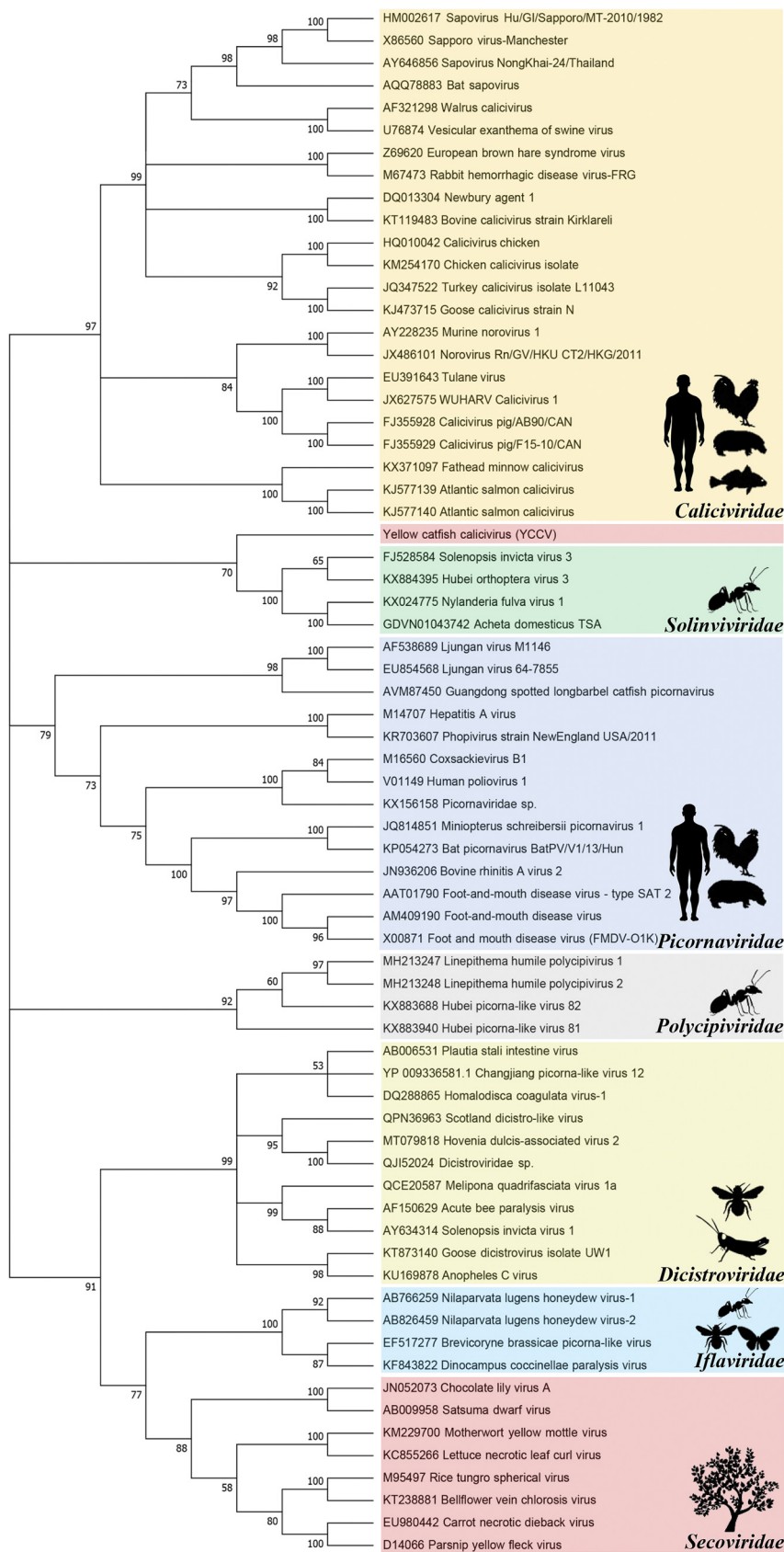

**FIG 6** Phylogenetic analysis based on amino acid sequences of the nonstructural protein from the order *Picornavirales*. Seven representative families in the order *Picornavirales* were used to construct

noro-, reco-, and vesiviruses, there is also a minor structural capsid protein, VP2, encoded by a separate ORF (22–24). YcCV NS1 had ~27.23% amino acid similarity with the NS proteins from other caliciviruses, and in the phylogenetic analysis, it formed a distinct branch close to other caliciviruses, which supported our finding that a novel calicivirus, YcCV, infects yellow catfish. This is not the first calicivirus to be identified from teleost fish species. The genera *Minovirus* (fathead minnow calicivirus [FMCV]) and *Salovirus* (Atlantic salmon calicivirus [ASCV]) in *Caliciviridae* have been shown to infect fish (25, 26). However, alignments of nucleotide and amino acid sequences of YcCV by BLAST showed no significant matches with these two viruses. Moreover, the genome organization of YcCV is composed of three ORFs, in contrast to FMCV and ASCV, which have two ORFs, although their total genome sizes (7.4 kb) are similar. These results support our finding that a novel calicivirus, YcCV, could infect yellow catfish.

The phylogenetic proximity of the amino acid sequences of YcCV and the members of *Caliciviridae* was relatively low based on the NS protein (~27.23%); moreover, for the *Caliciviridae*, there are no accepted criteria to assign genus and species names. Recently, dual nomenclature employing both NS and VP1 sequences was proposed according to the relatively common occurrence of recombination among these viruses; thus, recognition of recombinants might be relevant (27). Considering this proposal and the variation in intergenus sequence similarity among caliciviruses (including YcCV), we propose that YcCV be tentatively termed *Yecavirus*, as a prototype virus of a novel genus of *Caliviviridae*.

Many pathogens are recognized as challenges to the health of cultured yellow catfish, including *Edwardsiella ictaluri*, *Aeromonas veronii*, *Vibrio mimicus*, *Streptococcus iniae*, white bream virus, etc. (3–5, 7, 28, 29). In our study, these pathogens were not detected in the naturally infected yellow catfish. In the practice of aquaculture, a bacterial agent(s) may conditionally infect fish with viral diseases. This agent(s) may also contribute to the mortality of virus-infected fish; thus, the bacterial agent(s) might be occasionally isolated from naturally diseased fish. However, in our study, no bacterial agent was isolated, and the challenge experiments using bacterium-free homogenates derived from the diseased fish tissues showed high pathogenicity of YcCV in the absence of bacteria. In addition, the main symptoms of yellow catfish infected with bacterial agents, such as *Aeromonas veronii*, etc., were abdominal enlargement and ascites in the abdominal cavity; however, no hemorrhages were observed on the body surface (5). The clinical symptoms are different from those of YcCV-infected fish. These results further confirmed that YcCV is the causative agent responsible for the specific clinical symptoms and massive mortality in farmed yellow catfish.

A wide range of mammals, as well as fish and birds, can harbor caliciviruses (16). Except for certain vesiviruses in the family *Caliciviridae*, caliciviruses generally show a natural host restriction. This represents a considerable challenge to the study of the biology of viruses from genera without available productive cell culture systems. In our study, eight cell lines (CCK, CCO, EPC, GCO, RTG-2, FHM, GiCB, and CrEK) were tested for their susceptibility to YcCV. The results showed that YcCV could propagate in CCK cells; however, the cytopathic effect (CPE) in CCK cells gradually became inapparent in successive passages. To date, no cell line derived from yellow catfish is available for the isolation of YcCV. Thus, there is an urgent need to establish a highly susceptible cell line from yellow catfish for the investigation of the infection mechanism of YcCV and the development of methods for the prevention and control of this disease.

The identification and analysis of virus infection are frequently performed using PCR. In this study, we developed an RT-PCR method to detect YcCV in diseased yellow

**FIG 6** Legend (Continued)
the phylogenetic tree, using the maximum-likelihood method with 500 bootstrap replicates. The novel YcCV is indicated by pink shading, and the numbers under the branches indicate the bootstrap values (values lower than 50 are hidden). The genetic distance (as the number of substitutions per site) is represented by the scale bar.

catfish and implemented it for epidemiological studies. The method has potential for rapid and safe RT-PCR-based diagnosis of YcCV infection of yellow catfish in the future. In addition, a riboprobe to detect the YcCV NS gene sequence was designed and used in *in situ* hybridization, demonstrating its specificity in the analysis of RNA viruses, which is capable of detecting YcCV sensitively.

In conclusion, a novel virus was identified as the causative agent of a severe emerging disease in farmed yellow catfish in China. The virus is a member of the family *Caliciviridae* and was tentatively named yellow catfish calicivirus (YcCV) based on its morphology, sequence characterization, phylogenetic analysis, FISH analysis, and pathogen challenge experiments. Screening of a larger number of samples in China might help to further confirm the high prevalence of YcCV in farmed yellow catfish. Further research is required to advance efforts in investigating the infection mechanism of the virus and developing techniques for the control of the disease.

## MATERIALS AND METHODS

**Fish.** Diseased yellow catfish (mean weight, 25 g) showing hemorrhage on the jaw were collected from a commercial yellow catfish farm in Qianjiang City, Hubei Province, China, in April 2020. The diseased fish were transferred alive to the laboratory for detailed diagnosis and determination of the pathogen. Healthy yellow catfish (10 to 13 cm in length) for animal experiments were obtained from a yellow catfish breeding farm at Lu Lake in Jiangxia, Wuhan, Hubei Province. This yellow catfish breeding farm had no recorded history of this disease. Recirculation rearing systems (2.1 m by 1.6 m by 1.6 m) with aerated water at 25°C were used to acclimate the healthy fish, which were fed with commercial diets once a day for 1 week prior to experimental infection.

**Bacteriology and parasitology.** The viscera, gills, and exterior mucus of diseased yellow catfish with typical clinical signs were sampled and examined for parasites under a light microscope. Moribund fish were sacrificed humanely, and their kidneys, spleens, and livers were inoculated onto plates containing brain heart infusion (Difco, Franklin Lakes, NJ, USA) agar and incubated for 10 days at 25°C for bacterial isolation.

**Histopathology.** Tissue samples from naturally diseased moribund yellow catfish were collected for histological observation by using hematoxylin and eosin (H&E) staining, according to a previously described protocol (30). Tissues were dissected and fixed for 24 h in 4% paraformaldehyde (PFA) at 4°C, and then rinsed using Dulbecco's phosphate-buffered saline (DPBS; Sigma, St. Louis, MO, USA). After dehydration through a graded ethanol series to absolute ethanol, optimum cutting temperature compound (OCT) was used to embed the samples, which were then cut using a cryostat (CM1950; Leica, Germany) into 8-$\mu$m-thick sections at −20°C. The slices were stained with H&E and observed under a light microscope (DM2500; Leica).

**Electron microscopy.** Intestine, gill, brain, heart, kidney, spleen, and liver tissues collected from moribund yellow catfish were fixed in 2.5% glutaraldehyde, postfixed in aqueous 1% osmium tetroxide ($OsO_4$) for 1 h after washing with DPBS, dehydrated in a graded series of ethanol, embedded in Epon epoxy resin, sectioned, and stained using lead citrate and 2% uranyl acetate as described previously (31). A transmission electron microscope (Hitachi-7650; Hitachi, Tokyo, Japan) was then used to observe the samples at 80 kV. YcCV-infected CCK cells were fixed using 2.5% glutaraldehyde overnight at 4°C. Next day, the cells were removed from the flask by scraping and subjected to centrifugation for 10 min at $1,000 \times g$. The cell pellet was subjected to electron microscopic examination as described above.

**RNA-seq analysis.** Kidney and spleen tissues from diseased yellow catfish were collected for RNA-seq analysis. A TRIzol reagent kit (Invitrogen, Carlsbad, CA, USA) was used to extract total RNA following the manufacturer's guidelines. An Agilent 2100 Bioanalyzer (Agilent Technologies, Palo Alto, CA, USA) was used to quantify the total RNA, and its quality was assessed via RNase-free agarose gel electrophoresis. rRNAs were removed from the extracted total RNA, leaving mRNAs and noncoding RNAs (ncRNAs). Fragmentation buffer was used to produce short fragments of the ncRNAs and mRNAs, which were then subjected to reverse transcription to cDNAs using random primers. A reaction mixture comprising DNA polymerase I, RNase H, a deoxynucleoside triphosphate (dNTP) (dUTP instead of dTTP), and buffer was used to produce second-strand cDNA. Next, a QIAquick PCR extraction kit (Qiagen, Venlo, the Netherlands) was used to purify the cDNA fragments, which were end repaired, given poly(A) tails, and ligated with Illumina (San Diego, CA, USA) sequencing adapters. Subsequently, the second-strand cDNA was digested using uracil-*N*-glycosylase (UNG). Agarose gel electrophoresis was then used to size select the digested products, followed by PCR amplification and sequencing by Gene Denovo Biotechnology Co. (Guangzhou, China) using an Illumina HiSeq TM 4000 instrument. Bowtie 2 (v2.0.6) was used to map the filtered reads against the yellow catfish reference genomes to remove the host sequences. The MIRA assembler was then used to *de novo* assemble the remaining reads, and similarity searches of contigs and unique singletons that showed little or no similarity at the nucleotide level were carried out using BLASTx against the GenBank protein database. The contigs were compared to the complete viral RefSeq database using BLASTx and BLASTp in NCBI. The final contigs were annotated using Geneious (v. 9.1.3; Biomatters, Auckland, New Zealand) (32, 33).

**Virus isolation.** For virus isolation, eight fish cell lines were used for virus susceptibility tests, including channel catfish kidney (CCK), channel catfish ovary (CCO), epithelioma papilloma cyprinid (EPC),

**TABLE 1** Primers used in present study

| Primer name | Sequence (5′→3′) | Primer position in YcCV (bp) | Purpose |
|---|---|---|---|
| YcCV-rF1 | CGGGAACGGTAACTGAGTTGATTGCTGC | 7082–7109 | RACE-PCR to amplify the 3′ and 5′ termini |
| YcCV-rF2 | CGGATACCTTTAATGCGTGTGTCCACCC | 7172–7199 | RACE-PCR to amplify the 3′ and 5′ termini |
| YcCV-rR1 | TCGCCGCCAACCGTCCACAACCCTTTCCA | 731–759 | RACE-PCR to amplify the 3′ and 5′ termini |
| YcCV-rR2 | GAAAGCCAGGTGGCAAGCCTGAGGTGTG | 1931–1958 | RACE-PCR to amplify the 3′ and 5′ termini |
| YcCV-F1 | TCTGGAAGACACCCTAGCCAAAGCA | 455–479 | Overlap RT-PCR to amplify the genome of YcCV |
| YcCV-R1 | ACGCACCCACCCCGAGAATATGAAA | 1667–1691 | Overlap RT-PCR to amplify the genome of YcCV |
| YcCV-F2 | GTCATCACATCCAATCACCTACCCA | 1467–1491 | Overlap RT-PCR to amplify the genome of YcCV |
| YcCV-R2 | ACCGACAACACTACCAAACCAAAAT | 2867–2891 | Overlap RT-PCR to amplify the genome of YcCV |
| YcCV-F3 | GTCTGTGTGGATGATGAGTGCTGTC | 2127–2151 | Overlap RT-PCR to amplify the genome of YcCV |
| YcCV-R3 | ATGTGGCTTGTCTTGTTGTTGCTAG | 3601–3625 | Overlap RT-PCR to amplify the genome of YcCV |
| YcCV-F4 | TGGCTCGCTTCTACTTTCAGATTGC | 3365–3389 | Overlap RT-PCR to amplify the genome of YcCV |
| YcCV-R4 | AAAAGCGCGCGAACTGGAAATAGGT | 4935–4959 | Overlap RT-PCR to amplify the genome of YcCV |
| YcCV-F5 | AACAAGCACCGACTGGAGCAAAATC | 4229–4253 | Overlap RT-PCR to amplify the genome of YcCV |
| YcCV-R5 | GGGGGGTCAAAAAACCGTTAAAGGA | 5517–5541 | Overlap RT-PCR to amplify the genome of YcCV |
| YcCV-F6 | TCGTCTGAGTTGACCTGTTGGGGATT | 5259–5283 | Overlap RT-PCR to amplify the genome of YcCV |
| YcCV-R6 | CCTTTGGTGACAAGCCTTGAGAACA | 7285–7309 | Overlap RT-PCR to amplify the genome of YcCV |
| YcCV-F | CGCCTAAAGTTCTCCTCCTTGTTGG | 1147–1171 | RT-PCR to detect the NS gene of YcCV |
| YcCV-R | AAAGTTGGGTGGATTGGTTTGTCAT | 1645–1669 | RT-PCR to detect the NS gene of YcCV |
| YcCV-Fq | GCCTAAAGTTCTCCTCCTTGTTG | 1148–1170 | qRT-PCR to detect YcCV |
| YcCV-Rq | AGCAGCTCAGCATTTAATTCATC | 1359–1381 | qRT-PCR to detect YcCV |
| YcCV-Fi | GTCTTACGGGTCCGAGTTTTCCTGGAACGA | 4639–4668 | ISH to detect YcCV RNA expression in infected yellow catfish |

grass carp ovary (GCO), rainbow trout gonadal (RTG-2), fathead minnow (FHM), gibel carp brain (GiCB), and Chinese rice field eel kidney (CrEK) cell lines. These cells were cultured in T-25 (25-cm²) flasks (Corning Inc., Corning NY, USA) in medium 199 (M199) (Sigma-Aldrich, St. Louis, MO, USA) with 10% fetal bovine serum (FBS) at 20°C overnight before infection. Spleen and kidney tissues sampled from diseased yellow catfish were homogenized in DPBS (tissue-DPBS ratio, 1:10), and then freeze-thawed for three cycles (from −80°C to room temperature), followed by centrifugation for 20 min at 4,000 × *g* and 4°C (Sigma-3K15). The supernatant was collected and filtered through a 0.22-$\mu$m filter (Nalgene, Rochester, NY, USA), and finally, 1 mL of filtrate at 1:10, 1:100, and 1:1,000 dilutions was inoculated onto confluent cell monolayers for each T-25 flask. The negative-control (mock-infected) cells were inoculated with the same volume of M199 medium. The filtrate was allowed to absorb cells for 1 h, and then 4 mL of medium containing 2% FBS was added to the flasks. The cell cultures were incubated at 20°C and checked daily under an inverted phase-contrast microscope (Nikon, Tokyo, Japan) to observe the cytopathic effect (CPE). The supernatant of cells showing CPE was collected, and YcCV was detected by RNA extraction followed by RT-PCR. If no CPE was observed, the culture was subjected to blind passaging for three generations, after which the sample was considered negative for virus isolation (31).

**RT-PCR detection.** The TRIzol reagent (Invitrogen) was used to extract RNA from the kidneys of diseased yellow catfish following the manufacturer's guidelines. Reverse transcription was performed using a PrimeScript first-strand cDNA synthesis kit (TaKaRa, Shiga, Japan). The partial genomic sequence of YcCV from the RNA-seq analysis was used to design RT-PCR primers (YcCV-F/R) (Table 1), and an RT-PCR amplification kit (TaKaRa) was used to perform the RT-PCRs. Subsequently, the cDNA samples were subjected to PCR (35 cycles of 95°C for 5 min, 94°C for 1 min, and 58°C for 1 min, and 72°C for 1 min, followed by extension at 72°C for 10 min), which produced a 523-bp PCR product.

**Complete viral genome sequencing.** Overlapping RT-PCR was carried out using designed pairs of primers (YcCV-F1–YcCV-R1 to YcCV-F6–YcCV-R6) (Table 1) to obtain the sequence of YcCV (31). A Clontech SMART cDNA synthesis kit (TaKaRa) was used to perform rapid amplification of cDNA ends (RACE) PCR with the aim of identifying the 5′ and 3′ ends of YcCV from diseased yellow catfish. The 5′ region of the cDNA sequence was extended using 5′ RACE with a gene-specific primer (YcCV-rR1–YcCV-rR2) (Table 1). A TaKaRa RNA PCR kit (TaKaRa) was used to carry out 3′ RACE using an oligo(dT) adapter primer and YcCV-rF1–YcCV-rF2 (Table 1). Agarose gel electrophoresis was used to check the PCR product, and then the product was purified using the Wizard SV gel and PCR clean-up system (Promega, Madison, WI, USA) and ligated into vector pMD 19-T (TaKaRa, Japan) at 4°C overnight for cloning and sequencing. The primers listed in Table 1 were used for RT-PCR and RACE to detect the virus.

**Quantitative real-time qRT-PCR.** Samples (5 ± 0.10 mg) of intestine, gill, brain, heart, kidney, spleen, and liver were collected from fish and homogenized in lysis buffer. Total RNA of YcCV were extracted using TRIzol reagent (Invitrogen, USA), according to the manufacturer's instructions. The RNA was reverse transcribed to cDNA using a PrimeScript first-strand cDNA synthesis kit (TaKaRa, Japan). The copy numbers of the YcCV in different tissues were examined using qPCR with normalized cDNA as the template. A SYBR green real-time PCR mix (ToYoBo, Osaka, Japan) and the Rotor-Gene Q real-time PCR detection system (Qiagen, Hilden, Germany) were used to perform qPCR following the manufacturer's protocols. The qPCR cycle conditions were 95°C for 10 min followed by 40 cycles of 95°C for 10 s, 60°C

for 1 min, and 72°C for 30 s, using the primers (YcCV-Fq–YcCV-Rq) listed in Table 1. The experiment was performed three times independently.

**Prevalence of YcCV infection among yellow catfish.** To determine the prevalence of YcCV, 172 diseased yellow catfish were collected from 10 locations in Hubei (Qianjiang, Jingmen, Yichang, Wuhan, Honghu, Jiayu, Xiantao, Zhijiang, and Jingzhou) and Sichuan (Leshan), China. Samples were collected between March and June during 2020 and 2021. The spleens and kidneys were sampled and subjected to RT-PCR analysis to detect YcCV (Table S1; Fig. 4G). The TRIzol reagent (Invitrogen) was used to extract RNA from the tissues following the manufacturer's guidelines; then the RNA was converted to cDNA by RT-PCR. For positive samples, the NS gene was amplified using primers YcCV-F and YcCV-R and sequenced.

**FISH.** FISH was applied to determine YcCV RNA expression in infected yellow catfish by using a digoxigenin (DIG)-labeled probe that binds to the YcCV nonstructural protein sequence (GenBank accession no. MZ065194), which was synthesized as described previously (34). Table 1 showed the YcCV-Fi primer sequences. Infected tissues from yellow catfish were fixed using 4% PFA (Sigma) overnight at 4°C. The tissues were then washed with PBS three times and embedded in OCT (Leica), and 8-$\mu$m sections were acquired at $-20$°C (Leica CM1950). The sections were then labeled using the DIG-labeled probe at 65°C overnight. The tissues were washed for 30 min each time using 2$\times$ SSC (1$\times$ SSC is 0.015 M sodium citrate and 0.15 M sodium chloride and) followed by 0.2$\times$ SSC, washed with PBS for 10 min, and incubated for 3 h in 0.1% bovine serum albumin (BSA) in PBS. Next, anti-DIG antibodies (1:4,000; Roche, Mannheim, Germany) were added and incubated with the cells at 4°C overnight, before staining with 4',6-diamidino-2-phenylindole (DAPI) (1 $\mu$g/mL) for 5 min. A light microscope was used to examine the tissues (DM2500; Leica, Germany).

**Phylogenetic analysis.** Homology comparison was conducted using the full-length sequence of YcCV using BLASTn and BLASTp (NCBI database). The Expert Protein Analysis System was used to analyze the deduced amino acid sequences. The Muscle package with default parameters was used to carry out the multiple sequence alignments. MEGA 6.0 was used to construct a phylogenetic tree based on nonstructural protein sequences from other members of the order of *Picornavirales* using the maximum-likelihood method combined with the Poisson substitution model and rates of gamma distribution. Bootstrapping was performed with 500 replicates.

**Animal experiments.** Healthy yellow catfish were used for challenge experiments. The fish were randomly divided into an experimental group and a control group (30 per group). The fish in the experimental group were challenged by intraperitoneal (i.p.) injection of 0.5 mL filtrate obtained through 0.22-$\mu$m filters from diseased fish tissue homogenates with a titer of $10^{7.0}$ copies/mL. Fish in the control group were injected with 0.5 mL of DPBS i.p. During the infection experiment, all fish were reared in tanks with aerated water at 20°C. The clinical signs and mortality of the fish were monitored and recorded daily. Three moribund or dead fish per group (including controls) were collected randomly, and their tissues were processed for RT-PCR to detect presence or absence of the virus.

**Ethics statements.** The study was carried out strictly in accordance with the Guide for the Care and Use of Laboratory Animals of the Monitoring Committee of Hubei Province, China, and the protocol was approved by the Committee on the Ethics of Animal Experiments at the Yangtze River Fisheries Research Institute, Chinese Academy of Fishery Sciences (approval number YFI2020-03). Yellow catfish were euthanized by administration of 1 mg/mL of MS-222 (Sigma, USA) for 20 to 30 min before tissue collection.

**Data availability.** The YcCV genomic sequence is available in the NCBI GenBank database (accession number MZ065194).

## SUPPLEMENTAL MATERIAL

Supplemental material is available online only.
**SUPPLEMENTAL FILE 1**, PDF file, 0.4 MB.

## ACKNOWLEDGMENTS

This work was supported by the National Key R&D Program of China (grant 2019YFD0900102), the Central Public-interest Scientific Institution Basal Research Fund, CAFS (grants 2021JBF04 and 2020XT08), and the China Agriculture Research System of MOF and MARA (grant CARS-45-16).

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
