## [Reviewer comments · Microbiology Spectrum]

Microbiology Spectrum

A novel RNA virus causing massive mortality in yellow catfish, *Pelteobagrus fulvidraco*, was characterized as an emerging genus in *Caliciviridae*, *Picornavirales*

Wenzhi Liu, Mingyang Xue, Tao Yang, Yiqun Li, Nan Jiang, Yuding Fan, Yan Meng, Xiaowen Luo, Yong Zhou, and Lingbing Zeng

Corresponding Author(s): Lingbing Zeng, Yangtze River Fisheries Research Institute

Review Timeline:

Submission Date:	February 18, 2022
Editorial Decision:	May 24, 2022
Revision Received:	July 8, 2022
Accepted:	July 17, 2022

Editor: Frederick S. Kibenge

Reviewer(s): The reviewers have opted to remain anonymous.

Transaction Report:

DOI: <https://doi.org/10.1128/spectrum.00624-22>

May 24, 2022

Prof. Lingbing Zeng
Yangtze River Fisheries Research Institute
Division of Fish Disease
No. 8, 1st Wudayuan Road
East Lake Hi-Tech Development Zone
Wuhan, Hubei 430223
China

Re: Spectrum00624-22 (A novel RNA virus causing massive mortality in yellow catfish, *Pelteobagrus fulvidraco*, was characterized as an emerging genus in *Caliciviridae*, *Picornavirales*)

Dear Prof. Lingbing Zeng:

Your manuscript has been reviewed by two reviewers and both recommend modifications.

Link Not Available

Sincerely,

Frederick S. Kibenge

Journals Department
Reviewer comments:

Reviewer #2 (Comments for the Author):

In the manuscript by Liu et al., the authors isolated and characterized a novel virus from yellow catfish. The authors characterized the virus by electron microscopy and genomic cloning and sequencing. The genome sequence was deposited in the GenBank database. The authors further genetic characterized the sequence. Through the studies on viral morphology, genome organization, sequence homology, the authors identified the new isolated virus belongs to the family Caliciviridae. The authors further show that the newly isolated virus does in fact cause severe pathology in experimentally infected yellow catfish.

Overall, this is a very well presented and interesting manuscript. While the data presented supports the authors conclusions, the manuscript would be improved if the following concerns were addressed:

1, line 92-94. " Brian heart infusion agar plates were used for bacterial isolation from the liver and kidney of moribund fish, and no bacterial infection was confirmed."

line 92-94. "Genomic sequences of the host, plants, bacteria, parasites, and fungi were screened out before assembly of the transcripts. "

The two descriptions contradict each other.

2, RNA sequencing (RNA-seq) approach was performed using kidney and spleen tissues of diseased fish in this study. However, the presence of potential DNA viruses in tissues cannot be ruled out. DNA virus detection (For example, random primer detection for DNA viruses) should be performed.

3, YcCV could cause clear CPE in CCK cells. Thus, plaque assay maybe applicable to this virus. The plaque purified virus for animal experiments will be more convincing.

4, Samples from nine cities in two provinces showed positivity for YcCV. Please show the gene homology of those viruses coming from different regions.

Reviewer #3 (Comments for the Author):

A novel RNA virus causing huge economic loss in China was firstly identified from yellow catfish. The study was well designed; data could support the conclusion.

1. Please add the titer for the new virus and dosage that was used in the artificial challenge, The clinical signs caused in the artificial challenge should be included to make a comparison with the natural infection. however , there are a few minor revision should be made.
2. According to the phylogenetic tree, the new virus seems to be within the family Solinviviridae. Please highlight the reason why the new virus was not classified to this family.
3. The results of the prevalence investigation for the new virus was missing.

Staff Comments:

Preparing Revision Guidelines

Please return the manuscript within 60 days; if you cannot complete the modification within this time period, please contact me. If you do not wish to modify the manuscript and prefer to submit it to another journal, please notify me of your decision immediately so that the manuscript may be formally withdrawn from consideration by Microbiology Spectrum.

In the manuscript by Liu et al., the authors isolated and characterized a novel virus from yellow catfish. The authors characterized the virus by electron microscopy and genomic cloning and sequencing. The genome sequence was deposited in the GenBank database. The authors further genetically characterized the sequence. Through the studies on viral morphology, genome organization, sequence homology, the authors identified the new isolated virus belongs to the family Caliciviridae. The authors further show that the newly isolated virus does in fact cause severe pathology in experimentally infected yellow catfish. Overall, this is a very well presented and interesting manuscript. While the data presented supports the authors' conclusions, the manuscript would be improved if the following concerns were addressed:

1, line 92-94. "Brain heart infusion agar plates were used for bacterial isolation from the liver and kidney of moribund fish, and no bacterial infection was confirmed."

line 92-94. "Genomic sequences of the host, plants, bacteria, parasites, and fungi were screened out before assembly of the transcripts. "

The two descriptions contradict each other.

2, RNA sequencing (RNA-seq) approach was performed using kidney and spleen tissues of diseased fish in this study. However, the presence of potential DNA viruses in tissues cannot be ruled out. DNA virus detection (For example, random primer detection for DNA viruses) should be performed.

3, YcCV could cause clear CPE in CCK cells. Thus, plaque assay maybe applicable to this virus. The plaque purified virus for animal experiments will be more convincing.

4, Samples from nine cities in two provinces showed positivity for YcCV. Please show the gene homology of those viruses coming from different regions.

Response to Reviewers (Spectrum00624-22)

Dear Editors:

Thank you very much for your letter and the reviewer's comments on our manuscript entitled "A novel RNA virus causing massive mortality in yellow catfish, *Pelteobagrus fulvidraco*, was characterized as an emerging genus in *Caliciviridae*, *Picornavirales*" (Spectrum00624-22). We have made every effort to incorporate the reviewer's suggestions into a revised manuscript and feel it greatly improved. The comments provided are greatly appreciated. Responses to reviewer's comments and details regarding the revisions are described as following:

Reviewer #2:

In the manuscript by Liu et al., the authors isolated and characterized a novel virus from yellow catfish. The authors characterized the virus by electron microscopy and genomic cloning and sequencing. The genome sequence was deposited in the GenBank database. The authors further genetic characterized the sequence. Through the studies on viral morphology, genome organization, sequence homology, the authors identified the new isolated virus belongs to the family *Caliciviridae*. The authors further show that the newly isolated virus does in fact cause severe pathology in experimentally infected yellow catfish. Overall, this is a very well presented and interesting manuscript. While the data presented supports the authors conclusions, the manuscript would be improved if the following concerns were addressed:

1. line 92-94. " Brian heart infusion agar plates were used for bacterial isolation from the liver and kidney of moribund fish, and no bacterial infection was confirmed." line 92-94. "Genomic sequences of the host, plants, bacteria, parasites, and fungi were screened out before assembly of the transcripts. " The two descriptions contradict each other.

Response: Thank you very much for the comments and suggestions. In our study, the causative pathogen was characterized in diseased yellow catfish by using clinical observation, electron microscopy, parasite observation, bacterial identification,

RNA-seq, sequence characterization, reverse transcription PCR (RT-PCR), animal experiments, et al. Bacterial isolation was attempted from the kidney, spleen and liver of moribund fish on BHI agar plates, and no bacterial infection was confirmed. By transmission electron microscopy observation, a large number of spherical viral particles were revealed in the cytoplasm of cells in kidney and spleen of diseased fish. The virus particle was approximately 35 nm in diameter. So, an RNA sequencing (RNA-seq) approach was performed using kidney and spleen tissues of diseased fish to identify and characterize this virus. Using this approach, *in silico* translation of the transcripts provides protein sequences from diseased yellow catfish that can be compared with all known pathogenic sequences in database including some non-specific short sequences. These non-specific short sequences can often be aligned with some bacteria, fungi, plants, hosts, etc. which were actually not the causative pathogens. Therefore, we want to expressed that non-specific short genomic sequences of the host, plants, bacteria, parasites, and fungi were screened out before assembly of the transcripts. We apologize for the lack of clarity of the statement and make reviewer confused. We have revised to clarify this sentence in our revised manuscript (line 108).

2. RNA sequencing (RNA-seq) approach was performed using kidney and spleen tissues of diseased fish in this study. However, the presence of potential DNA viruses in tissues cannot be ruled out. DNA virus detection (For example, random primer detection for DNA viruses) should be performed.

Response: Thank you for this suggestion. In our study, RNA sequencing approach was performed to obtain the causative pathogen in diseased yellow catfish. In this method, RNA (including the mRNA) is amplified randomly and subjected to deep sequencing to generate transcript data. These transcript data were compared with all known pathogenic sequences in database to obtain the causative pathogen in diseased yellow catfish. If potential DNA virus was the causative pathogen of diseased yellow catfish in our study, it can replicate and transcribe viral mRNA in diseased tissues of yellow catfish. Thus, DNA virus could also be detected by RNA-seq approach. Therefore, RNA-seq approach could be performed to detect both

RNA and DNA virus in diseased yellow catfish if they were the causative pathogen.

3. YcCV could cause clear CPE in CCK cells. Thus, plaque assay maybe applicable to this virus. The plaque purified virus for animal experiments will be more convincing.

Response: Many thanks. In our study, eight cell lines (CCK, CCO, EPC, GCO, RTG-2, FHM, GiCB, and CrEK) were tested for their susceptibility to YcCV. The results showed that YcCV could propagate in CCK cells after three consecutive blind passages of virus culture; however, the cytopathic effect (CPE) in CCK cells gradually became inapparent in following successive passages. To date, no cell line derived from yellow catfish and CCK cell line is available for the isolation of YcCV in successive passages. So, it is a little bit difficult that the plaque purified virus from CCK cell line for animal experiments were conducted in our study. Of course, this is very good suggestion and a choice in future for further investigating this test in yellow catfish. In addition, healthy yellow catfish were challenged with bacteria-free tissue homogenates derived from naturally YcCV-infected yellow catfish showing hemorrhagic symptoms and mortality was as high as 90% at 10 dpi in our study. The clinical system in animal experiment was similar with the natural infection of YcCV in yellow catfish. This result could also clearly verify YcCV as the etiological agent responsible for the emerging disease in yellow catfish farms.

4. Samples from nine cities in two provinces showed positivity for YcCV. Please show the gene homology of those viruses coming from different regions.

Response: Thank you very much for the suggestions. We have added the details to the parts of results in our revised manuscript. Results showed that samples were positive for YcCV in two provinces, nine cities during our sample collections, including the province Hubei (Qianjiang, Jingmen, Honghu, Jiayu, Wuhan, Xiantao, Yichang, Zhijiang, and Jingzhou) and Sichuan province (Leshan city). The whole viral genome sequence from nine cities were sequenced and the gene homology were 94.6%-99.2% nucleotide identity with each other (line 199-200).

Reviewer #3:

A novel RNA virus causing huge economic loss in China was firstly identified

from yellow catfish. The study was well designed; data could support the conclusion.

Response: Thanks for the reviewer's affirmation of our study for the first description of a novel pathogen in yellow catfish. Indeed, yellow catfish culture is an economically important freshwater species in China and other Asian countries. The present study described that such emerging disease in farmed yellow catfish had recently spread in main breeding areas in China, and this lethal disease is highly contagious and threatens to the fish farming industry. Therefore, determination and characterization of this virus (YcCV) is important and will assist in developing approaches to control or prevent YcCV disease from yellow catfish.

1. Please add the titer for the new virus and dosage that was used in the artificial challenge, The clinical signs caused in the artificial challenge should be included to make a comparison with the natural infection. however, there are a few minor revision should be made.

Response: Thanks for the reviewer's comments. We have added the titer of $10^{7.0}$ copies/ ml (by qPCR) for the novel virus and dosage of 0.5 ml/fish that was used in the artificial challenge in our revised manuscript. This titer of novel virus was detected from the homogenate (kidney and spleen) of diseased fish tissues with the titer of $10^{5.0}$ copies/ mg by using qPCR. One-gram tissues of kidney and spleen from diseased fish was collected and homogenized with 10 mL DPBS on ice, then was filtrated through 0.22 μ m filter before used for challenge test. In addition, the clinical signs caused in the artificial challenge have been included in revised manuscript when compared with the natural infection of diseased yellow catfish. This has been revised as “Healthy yellow catfish were challenged with bacteria-free tissue homogenates derived from naturally YcCV-infected yellow catfish mainly showing hemorrhagic symptoms that were similar to those found in naturally diseased fish” (line 458-460; line 176-177).

2. According to the phylogenetic tree, the new virus seems to be within the family Soliniviridae. Please highlight the reason why the new virus was not classified to this family.

Response: Many thanks to the reviewer's comments. The phylogenetic analysis

showed that YcCV was close to the *Soliniviridae* and *Caliciviridae* families, and closer to the *Nyfulvavirus* in the *Soliniviridae* family; however, the YcCV NS gene was found to possess ~27.23% aa similarity to other caliciviruses, but no similarity to the viruses in the *Soliniviridae* using NCBI BLASTP analysis. In addition, monophyly of the *Soliniviridae* within the larger picorna/calici-like group is not completely certain and it is possible that the *Soliniviridae* form a sister group to the *Caliciviridae*, although the phylogenetic clustering is inconclusive because of the difficulty in resolving tree topologies at this depth. In our study, the viral morphology, viral particle size, viral nucleic acid length, genome organization, and host range, suggest that YcCV might belong to a new genus in the family *Caliciviridae*.

3. The results of the prevalence investigation for the new virus was missing.

Response: Thank you very much for the suggestions. We are sorry about that we did not give a detailed explaining of the prevalence investigation for the new virus. Results showed that samples were positive for YcCV in two provinces, nine cities during our sample collections, including the province Hubei (Qianjiang (14/32, 43.8%), Jingmen (5/18, 27.8%), Jiayu (3/12, 25.0%), Wuhan (5/21, 23.8%), Xiantao (8/23, 34.8%), Yichang (4/16, 25.0%), Zhijiang (10/21, 47.6%) and Jingzhou (2/17, 11.8%)) and Sichuan province (Leshan (4/12, 33.3%)) (line 196-199). In addition, this disease breaks in more and more provinces in China diagnosed by clinical signs in this year. It is urgent to conduct more further investigations on this virus and disease.

With appreciating all the comments and suggestions raised by reviewers, we have revised the manuscript following these comments. We feel that the revised manuscript is improved much and presents the significance of the work in this study. The changes are marked in the marked-up paper. Again, we sincerely thank the editors and reviewers for their help with our manuscript.

July 17, 2022

Prof. Lingbing Zeng
Yangtze River Fisheries Research Institute
Division of Fish Disease
No. 8, 1st Wudayuan Road
East Lake Hi-Tech Development Zone
Wuhan, Hubei 430223
China

Re: Spectrum00624-22R1 (A novel RNA virus causing massive mortality in yellow catfish, *Pelteobagrus fulvidraco*, was characterized as an emerging genus in *Caliciviridae*, *Picornavirales*)

Dear Prof. Lingbing Zeng:

The authors have adequately addressed the reviewers' comments. The manuscript is accepted for publication.

Your manuscript has been accepted, and I am forwarding it to the ASM Journals Department for publication. You will be notified when your proofs are ready to be viewed.

Sincerely,

Frederick S. Kibenge
Editor, Microbiology Spectrum

The authors have adequately addressed the reviewers' comments. The manuscript is significantly improved. I recommend this manuscript be accepted for publication.